# Hybrid Dual-Attention Functional Map Learning for Robust Shape Matching

## Abstract

Although deep functional map methods have significantly advanced the field of 3D shape matching, many existing approaches still rely on conventional network architectures for feature enhancement and use only the Laplace-Beltrami operator (LBO) to construct eigenbases for functional map computation. This often results in performance degradation when the learned features are insufficiently distinctive. To overcome these limitations, we propose an efficient unsupervised framework for deformable shape matching. Our method incorporates a feature extraction module with a dual-layer attention mechanism, a differentiable functional map solver, and an optimal transport (OT) post-processing step to produce accurate point-to-point correspondences. The attention mechanism learns discriminative and structurally invariant descriptors, significantly improving robustness under complex geometric deformations. Additionally, we introduce a hybrid matching strategy that integrates both Laplacian and elastic modal representations, optimized via Sinkhorn iterations to yield a transport matrix. This facilitates robust and accurate correspondence recovery. Extensive experiments across diverse challenging scenarios demonstrate that our approach outperforms state-of-the-art methods in matching accuracy. Our code is publicly available.

## 1 Introduction

Shape matching is a long-standing and fundamental problem in computer graphics and shape analysis, with broad applications in areas such as texture transfer Dinh et al. (2005), deformation transfer Sumner & Popović (2004), pose transfer Song et al. (2023; 2021), segmentation Rustamov et al. (2007), reconstruction Leroy et al. (2024); Radford et al. (2021), and statistical shape analysis Egger et al. (2020); Li et al. (2017); Loper et al. (2023). Establishing accurate correspondences between non-rigid shapes remains highly challenging, especially under significant non-isometric deformations, partial occlusions, or topological inconsistencies. Early studies focused on handcrafted descriptors with geometric invariance-such as HKS Sun et al. (2009), WKS Aubry et al. (2011), and SHOT Salti et al. (2014)-to enable robust shape matching Aubry et al. (2011); Bronstein & Kokkinos (2010). Despite these efforts, such methods exhibit inherent limitations, and non-rigid shape matching continues to be an open and difficult problem Cao et al. (2023).

In recent years, spectral methods have gained growing interest for efficient shape modeling. A early prominent example is the functional map (FM) framework Ovsjanikov et al. (2012), which builds a mapping space using the Laplace-Beltrami operator (LBO) to encode shape features and establishes functional correspondences between shapes. A number of recent studies have enhanced the generality and robustness of the functional map estimation pipeline through the introduction of various regularization terms, robust loss functions, and effective post-processing techniques Huang et al. (2014); Rodolà et al. (2017); Sharma & Ovsjanikov (2020).

With the rise of deep learning, recent deep functional maps (DeepFM or DFM) have been widely adopted in various tasks Cao & Bernard (2023); Cao et al. (2023); Roetzer et al. (2024); Sun et al. (2023); Le et al. (2024). In contrast to classical approaches that rely on handcrafted features, these data-driven techniques learn feature representations by neural networks from training data. However, prevailing deep functional map methods primarily focus on estimating accurate functional correspondences; the subsequent conversion to point-to-point maps typically relies on nearest-neighbor

search or other post-processing operations. A significant limitation of this two-stage pipeline is its tendency to yield suboptimal point-wise matching accuracy.

On the other hand, most mainstream approaches rely on LBO eigenbases due to their solid theoretical foundation, well-understood eigenfunctions, and favorable mathematical properties. However, LBO eigenbases inherently emphasize low-frequency information, which improves global stability but often reduces sensitivity to high-frequency details—such as sharp creases and bends—thus compromising local matching accuracy Hartwig et al. (2023); Bastian et al. (2024).

To address these limitations, we proposed an efficient unsupervised shape matching framework, which contains an attention-based feature extraction module, a hybrid functional space module, and an efficient optimal transport-based post-processing step to establish precise point-to-point correspondences. The attention mechanism learns discriminative and structurally invariant descriptors, improving robustness under complex geometric deformations. The hybrid matching strategy integrates Laplacian and elastic spectral representations, which are optimized via Sinkhorn iterations to produce an approximately doubly-stochastic transport matrix, enabling accurate and robust correspondence estimation. Our main contributions can be summarized as follows:

- We introduce a dual-layer attention mechanism in the feature extraction module, which effectively learns discriminative and structurally consistent descriptors, enhancing the model's ability to adapt to complex geometric deformations.

- We adopt a hybrid matching strategy that integrates Laplacian and elastic modal representations, followed by Sinkhorn optimization. This approach yields an approximately doubly-stochastic transport matrix, from which high-quality point-to-point correspondences are robustly recovered.

- Extensive experiments across a variety of challenging scenarios demonstrate that the proposed method achieves substantial improvements in deformable shape matching performance.

## 2 BACKGROUND

### 2.1 (DEEP) FUNCTIONAL MAPS

The functional map method frames shape correspondence as a linear map between the spectral embeddings of two shapes Ovsjanikov et al. (2012). By leveraging compact matrices derived from truncated spectral bases, this mapping can be efficiently computed. Formally, let $\mathcal{X}$ and $\mathcal{Y}$ be two shapes (triangular meshes) with $n_x$ and $n_y$ vertices. By computing the first $k$ eigenfunctions of their LBO, the spectral bases can be represented as $\Phi_x \in \mathbb{R}^{n_x \times k}$ and $\Phi_y \in \mathbb{R}^{n_y \times k}$. Subsequently, given the geometric features $\mathcal{F}_x \in \mathbb{R}^{n_x \times d}$ and $\mathcal{F}_y \in \mathbb{R}^{n_y \times d}$ extracted from each shape, their projections onto the spectral bases yield the coefficient matrices $M = \Phi_x^\dagger \mathcal{F}_x$ and $N = \Phi_y^\dagger \mathcal{F}_y$, where $\dagger$ denotes the Moore–Penrose pseudoinverse and $d$ is the feature dimension. The functional map $Cxy$ is found by solving:

$$C_{xy} = \arg\min_C E_{data}(C) + \lambda E_{reg}(C). \tag{1}$$

The data term $E_{\text{data}}(C) = \|CM - N\|_F^2$ enforces the preservation of feature descriptors, while the regularization term $E_{\text{reg}}(C) = \|C\Lambda_1 - \Lambda_2 C\|_F^2$ imposes structural constraints on the mapping $C$. $\Lambda_1$ and $\Lambda_2$ denote the diagonal matrices of eigenvalues of the Laplace operators. Finally, a dense point-to-point map $\Pi_{yx} \in \{0,1\}^{n_y \times n_x}$ from the source to the target shape can be recovered using the relationship:

$$C_{xy} = \Phi_y^\dagger \Pi_{yx} \Phi_x C, \tag{2}$$

which is typically solved with a nearest-neighbor search or related techniques. However, traditional functional map methods rely heavily on the choice of initial descriptor functions, which may lack sufficient distinctiveness under complex deformations Roufosse et al. (2019). With the advancement of deep learning, the functional map framework has been extended into the Deep Functional Map (DFM) paradigm. The core idea is to learn a transformation of a given set of descriptors such that the optimal functional map derived from the transformed descriptors is as close as possible to a ground-truth correspondence provided during training. Unlike traditional functional maps that rely purely on the FM formulation, these deep learning-based approaches Roufosse et al. (2019); Litany

et al. (2017) typically employ an end-to-end loss function to supervise or constrain the learning of the functional map:

$$\min_{\hat{T}} \sum_{(\mathcal{X},\mathcal{Y}) \in \mathbb{G}} \ell_F\big(\text{Soft}(C), \text{GT}(\mathcal{X}, \mathcal{Y})\big), \qquad (3)$$

where $\hat{T}$ denotes a non-linear transformation (typically a neural network) applied to input descriptor functions; $\text{GT}(\mathcal{X}, \mathcal{Y})$ represents the set of training shape pairs for which ground-truth correspondences are available; and $\ell_F$ is a soft loss function that penalizes the deviation of the computed functional map after converting it into a soft correspondence matrix $\text{Soft}(C)$ from the ground truth.

## 2.2 ELASTIC EIGENMODES

The eigenfunctions of the LBO exhibit strong performance as a reduced functional basis in nearly isometric settings. However, they often fail to adequately align extrinsic shape features. To address this limitation, Hartwig et al. (2023) introduced a functional basis derived from the elastic eigenmodes of the shape's elastic energy, which improves extrinsic feature alignment. Specifically, the vibration modes of a shape $S$, are obtained from the Hessian of the total elastic energy evaluated at the identity, denoted as $\text{Hess} \, W_S[Id] \in \mathbb{R}^{3n \times 3n}$. This operator acts linearly on infinitesimal displacements $\psi_i \in (F(S))^3$ of the shape $S$, represented as vectors in $\mathbb{R}^{3n}$. The elastic eigenmodes are computed by solving the generalized eigenvalue problem:

$$\text{Hess} \, W_S[Id] \, \psi_i = \lambda_i A \psi_i. \qquad (4)$$

where $A$ is a block-diagonal matrix with stacked lumped mass matrices along the diagonal, and $\lambda$ denotes the eigenvalues. In practice, only the eigenvector $\psi_i$ associated with non-zero eigenvalues are retained. However, since these basis functions are not orthogonal, additional mathematical care is required in formulating the corresponding optimization problem. To address this issue, Bastian et al. (2024) proposed a joint spectral space combining LBO eigenfunctions and thin-shell Hessian energy basis functions, demonstrating that such a hybrid approach yields superior matching performance on both near-isometric and non-isometric shapes.

# 3 METHOD

Our work aims to robustly estimate functional maps and point-to-point correspondences for deformable shape pairs under diverse scenarios, including near-isometric and non-isometric deformations, as well as shapes exhibiting complex extrinsic geometric features. The core motivation of our method stems from the observation that existing deep functional map approaches typically rely on conventional network architectures for feature enhancement and use LBO-based eigenbases to compute functional maps. Such a framework tends to exhibit limited matching performance when features are insufficiently distinctive or when shapes contain extrinsic creases and high-curvature regions. To simultaneously enhance the accuracy of both functional maps and pointwise correspondences, we propose an efficient unsupervised framework that integrates a dual-layer attention-based feature extraction module, a hybrid functional space module, and an efficient optimal transport-based post-processing step to establish precise point-to-point correspondences. The overall framework is illustrated in Figure 1.

## 3.1 FEATURE EXTRACTION

To enhance the modeling of geometric structures and cross-shape information in non-rigid shape matching tasks, we design a feature extraction network incorporating a dual-layer attention mechanism. The architecture follows a Siamese design, where the inner layer is based on the DiffusionNet Sharp et al. (2022) backbone, and augmented with a Structure-Guided Channel Attention (SGCA) module to capture both local and global geometric features via spectral diffusion over the surface shapes.

The feature extractor is applied in a Siamese manner—using shared weights to process both shapes $\mathcal{X}$ and $\mathcal{Y}$—yielding initial pointwise features $F_X \in \mathbb{R}^{n_x \times d}$ and $F_Y \in \mathbb{R}^{n_y \times d}$, where $n_x$ and $n_y$

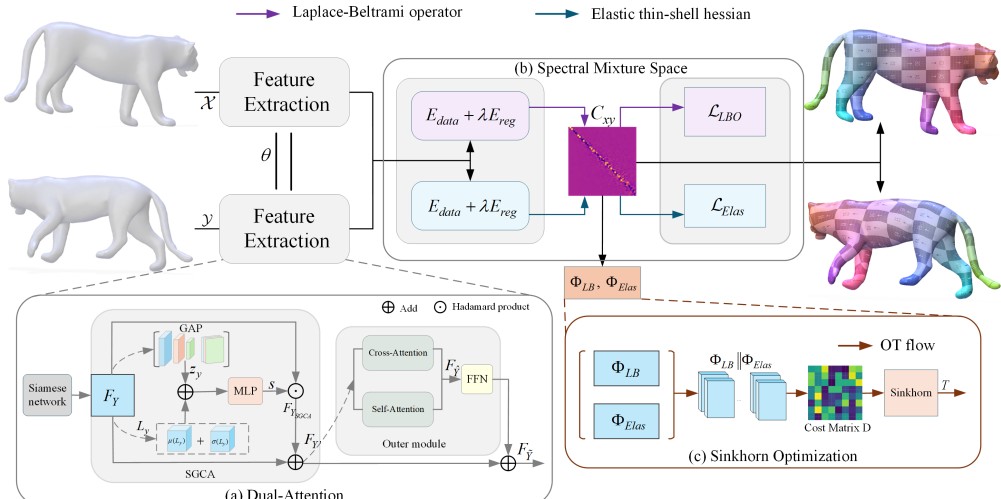

Figure 1: An efficient unsupervised shape matching framework. Given a pair of shapes $\mathcal{X}$ and $\mathcal{Y}$, we first employ a shared-weight feature extractor $\theta$ to obtain features $X$ and $Y$ (e.g., using the target shape features), where a dual-layer attention module (a) is integrated to enhance feature representation. Subsequently, these features are projected onto basis sets constructed from different linear operators. Then, within the constructed Spectral Mixture Space (b), we compute a block-diagonal functional map. Finally, in the post-processing stage, the Sinkhorn algorithm (c) is applied to refine the results, yielding high-quality point-to-point correspondences.

denote vertex counts and $d$ is the feature dimension. These features are then refined via the SGCA module. We compute channel statistics $z_x = GAP(F_x) \in \mathbb{R}^d$, $z_y = GAP(F_y) \in \mathbb{R}^d$, along with structure-aware vectors derived from the Laplacian operators: $g_x = [\mu(L_x), \sigma(L_x)] \in \mathbb{R}^2$ and $g_y = [\mu(L_y), \sigma(L_y)] \in \mathbb{R}^2$, where GAP denotes global average pooling. The channel and structure vectors are concatenated and passed through an MLP to generate attention weights: $s = \phi(W_2 \operatorname{ReLU}(W_1(z + \alpha \cdot g)))$, where $W_1 \in \mathbb{R}^{(d+2) \times d/r}$ and $W_2 \in \mathbb{R}^{d/r \in d}$, $\alpha$ is the structural guidance scaling factor, and $\phi$ represents the sigmoid activation function.

The attention weights are then expanded and applied element-wise to the original input feature: $F_{X_{\text{SGCA}}} = F_X \odot s$, $F_{Y_{\text{SGCA}}} = F_Y \odot s$. A residual connection is then applied to produce the final enhanced features: $F_{X'} = F_X + FX_{\text{SGCA}}$, $F_{Y'} = F_Y + FY_{\text{SGCA}}$.

The outer module enhances cross-shape feature alignment using a design inspired by the Predator network, incorporating residual connections and a Feed-Forward Network (FFN) Huang et al. (2021) to simulate cross-domain information transfer. It takes as input the features $F_{X'} \in \mathbb{R}^{n_x \times d}$ and $F_{Y'} \in \mathbb{R}^{n_y \times d}$ from the inner module. First, cross-attention is applied to the source features using the target features as key and value: $F_{\hat{X}} = \operatorname{Att}(F_{X'}, F_{Y'}, F_{Y'}) + F_{X'}$. This allows the source features to be enriched by information from the target. The same process is applied symmetrically to the target features: $F_{\hat{Y}} = \operatorname{Att}(F_{Y'}, F_{X'}, F_{X'}) + F_{Y'}$.

These bidirectional interactions promote balanced feature alignment between the shapes. Subsequently, self-attention refines the local structure and intra-shape consistency of both feature sets. Finally, the features are passed through an FFN with residual connections: $F_{\widetilde{X}} = FFN(F_{\hat{X}}) + F_{\hat{X}}$, $F_{\widetilde{Y}} = FFN(F_{\hat{Y}}) + F_{\hat{Y}}$. The resulting features retain local geometric information while incorporating cross-shape constraints, leading to more discriminative and stable representations for point-to-point matching.

## 3.2 SPECTRAL MIXTURE SPACE

To jointly capture low-frequency global alignment and high-frequency detail matching, we adopt the hybrid spectral strategy introduced in Bastian et al. (2024), which combines LBO eigenfunctions with elastic basis functions. Specifically, for each shape, the first $k$ orthogonal eigenfunctions $\Phi_k$ are obtained by solving the generalized eigenvalue problem $\Delta \phi_i = \lambda_i A \phi_i$, where $A$ is the mass matrix

encoding mesh area elements. These eigenfunctions provide stable low-frequency components that facilitate coarse global alignment while maintaining isometric invariance (we empirically set $k = 200$). Meanwhile, elastic basis functions $\psi$ are derived via the Hessian decomposition of the shape's elastic energy $W$, capturing high-frequency creases and fine local details. The LBO and elastic bases are combined into a hybrid-frequency space $\Phi_1, \ldots, \Phi_k, \psi_1, \ldots, \psi_{k_2}$. Within this spectral mixture space, the functional map $C$ is represented as a block matrix, where each sub-block $C_{ij}$ encodes the transformation between different types of basis functions.

Within this space, the functional map $C$ is represented as a block matrix:

$$C = \begin{pmatrix} C_{11} & C_{12} \\ C_{21} & C_{22} \end{pmatrix}, \tag{5}$$

where $C_{11}$ and $C_{22}$ correspond to intra-basis maps within the LBO and elastic bases, respectively, while $C_{12}$ and $C_{21}$ represent inter-basis mappings between them. This hybrid map integrates geometric consistency across candidate correspondences and is refined via the Sinkhorn optimal transport algorithm. The final point-to-point correspondence is determined by selecting matches with the highest probabilities, supporting either nearest-neighbor retrieval in the hybrid space or dense correspondence estimation. Further implementation details are described in the next section.

### 3.3 Sinkhorn Optimization

Inspired by Le et al. (2024), we introduce an efficient optimal transport mechanism for refining point-to-point correspondences. Integrated as a post-processing step during inference, the method enhances both accuracy and geometric consistency of the final maps. While traditional optimal transport can model complex structural alignment, it often suffers from high computational cost—$\mathcal{O}(n^2)$ in time and space—and limited performance in non-isometric settings. To improve robustness and geometric coherence, we incorporate multimodal features into the Sinkhorn algorithm. Specifically, we extract two sets of low-frequency functional embeddings: one from the Laplace–Beltrami (LB) spectral mapping, denoted $\Phi_{LB}^{\mathcal{X}}, \Phi_{LB}^{\mathcal{Y}}$, and another from the Elastic Modal Field mapping, denoted $\Phi_{elas}^{\mathcal{X}}, \Phi_{elas}^{\mathcal{Y}}$. These are concatenated to form unified geometric representations: $\Phi^{\mathcal{X}} = \left[\Phi_{LB}^{\mathcal{X}} \| \Phi_{elas}^{\mathcal{X}}\right], \Phi^{\mathcal{Y}} = \left[\Phi_{LB}^{\mathcal{Y}} \| \Phi_{elas}^{\mathcal{Y}}\right]$. A cost matrix $D$ is then defined using Euclidean distances between all point pairs: $D_{ij} = \|\Phi_i^{\mathcal{X}} - \Phi_j^{\mathcal{Y}}\|$. Based on the cost matrix, we formulate an entropy-regularized optimal transport objective as:

$$T' = \arg \min_{T \in \mathcal{U}(u, \nu)} \sum_{i,j} T_{ij} D_{ij} + \epsilon \cdot KL(T \| u \otimes \nu), \tag{6}$$

where $T$ is a doubly stochastic transport matrix, $\mathcal{U}(u, \nu)$ represents the set of joint distributions with marginals $u$ and $\nu$, $\epsilon$ is the regularization coefficient, and $KL(\cdot)$ refers to the Kullback–Leibler divergence. The cost term measures pairwise distances in the embedding space. To solve this optimization problem efficiently, we employ a log-domain implementation of the Sinkhorn algorithm. Specifically, we initialize $\log \alpha = -\frac{D}{\epsilon}$, and then iteratively apply a normalization in the log-space as $\log \alpha \leftarrow \log \alpha - \log \sum e^{\log \alpha_{ij}}$. The approximate transport matrix $T$ is recovered as: $T = \exp(\log \alpha)$, representing a soft probabilistic mapping from $\mathcal{Y}$ to $\mathcal{X}$. The final discrete correspondence is obtained by selecting the highest-probability match per column: $P(j) = \arg \max_i T_{ij}$, thus establishing point-to-point correspondence from target to source.

### 3.4 Loss function

To balance coarse global alignment with fine-grained local correspondence, we adopt a linear annealing strategy that progressively incorporates elastic basis features. During early training stages, the model relies primarily on functional descriptors derived from the LBO to establish coarse isometric correspondences, emphasizing global structural consistency. As training proceeds, elastic basis features are gradually introduced to enhance sensitivity to non-rigid deformations—such as bending and folding—and to capture local geometric details. The overall loss function is defined as a weighted combination:

$$\mathcal{L} = \mathcal{L}_{\text{LBO}} + \alpha \mathcal{L}_{\text{Elas}}, \tag{7}$$

where $\mathcal{L}_{\text{LBO}}$ denotes the isometric alignment loss in the LBO spectral space, and $\mathcal{L}_{\text{Elas}}$ measures non-isometric detail alignment using the elastic thin-shell basis. The weighting factor $\alpha$ increases linearly with training iterations.

# 4 EXPERIMENTAL RESULTS

**Baselines**. We compare our method with several representative non-rigid shape matching approaches, which are systematically categorized into the following groups: (1) axiomatic methods, including ZoomOut Melzi et al. (2019b), BCICP Ren et al. (2018), Smooth Shells Eisenberger et al. (2020), and DiscreteOp Ren et al. (2021); (2) supervised methods, such as FMNet Litany et al. (2017), GeomFMaps Donati et al. (2020), and TransMatch Trappolini et al. (2021); and (3) unsupervised methods including SURFMNet Roufosse et al. (2019), Deep Shells Eisenberger et al. (2020), ULRSSM Cao et al. (2023), Hybridfmap Bastian et al. (2024), and EOT Le et al. (2024). All selected baselines are relevant to our work and provide comprehensive comparisons.

**Metrics**. In accordance with all compared methods, we employ the average geodesic error ($\times 100\%$) as the primary evaluation metric for shape matching.

## 4.1 NEAR-ISOMETRIC SHAPE MATCHING

**Datasets**. We employ three widely-used benchmark datasets for near-isometric shape matching: FAUST Bogo et al. (2014), SCAPE Donati et al. (2022), and SHREC'19 Melzi et al. (2019a). To more rigorously assess algorithmic robustness under non-ideal conditions, we utilize remeshed versions of these datasets Cao et al. (2023), which present additional challenges compared to the original meshes. The FAUST dataset contains 100 human shapes (10 subjects in 10 different poses), with an 80/20 split between training and testing. The SCAPE dataset includes 71 different poses of the same individual, divided into 51 for training and 20 f or testing. The SHREC'19 dataset, a more challenging benchmark, includes 44 human shapes and is used exclusively for testing.

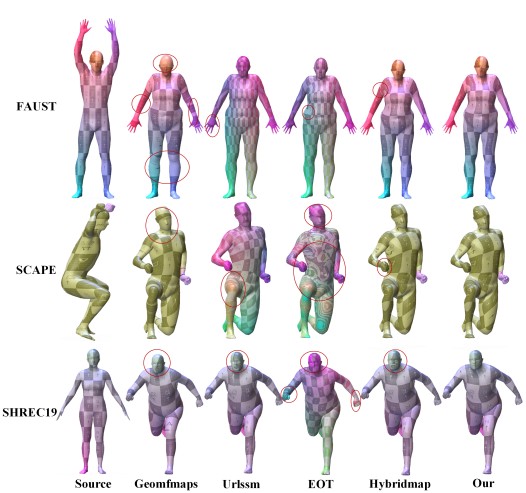

Figure 2: Qualitative results across different datasets. Correspondences are visualized via texture transfer. Regions with inaccurate mappings are highlighted with red circles.

**Results**. As summarized in Table 1, supervised methods generally achieve strong performance on their training data but are susceptible to overfitting, resulting in limited generalization to unseen shapes. In contrast, unsupervised approaches exhibit greater robustness and generalize more effectively across datasets. Our method outperforms recent alternatives such as Hybridfmap and EOT (an OT-based technique) in most settings, as shown quantitatively in Table 1. Moreover, qualitative results in Figures 3 demonstrate that our approach produces high-quality correspondences, comparable to or better than current state-of-the-art methods. The texture transfer visualizations in Figure 2 further illustrate the superiority of our method in terms of both alignment accuracy and smoothness of the correspondence field.

Table 2: Geo. error ($\times 100$) on SMAL dataset. Lower values indicate better correspondence accuracy.

| Method | Geo. error | Method | Geo. error |
|---|---|---|---|
| ZoomOut Melzi et al. (2019b) | 38.4 | DFAFMaps Luo et al. (2024) | 4.3 |
| Smooth Shells Eisenberger et al. (2020) | 36.1 | DRecovery Sundararaman et al. (2024) | 4.1 |
| DiscreteOp Ren et al. (2021) | 38.1 | RevisitingMap Cao et al. (2024b) | 3.6 |
| Hybrid Smooth Shells Bastian et al. (2024) | 28.4 | Hybridfmap Bastian et al. (2024) | 3.3 |
| MWP Deng et al. (2022) | 22.3 | Hybrid GeomFMaps Bastian et al. (2024) | 7.6 |
| FMNet Litany et al. (2017) | 42.0 | ULRSSM Cao et al. (2023) | 3.9 |
| GeomFMaps Donati et al. (2020) | 8.4 | SDUM Cao et al. (2024a) | **3.6** |
| EOT Le et al. (2024) | 4.4 | **Ours** | 4.3 |

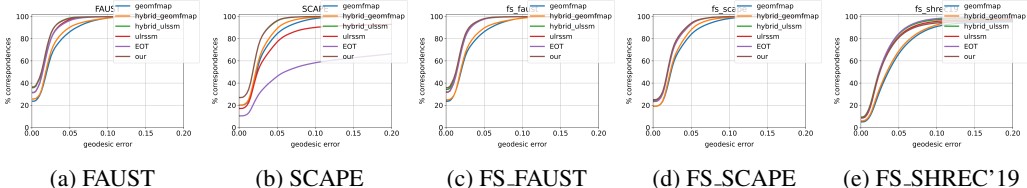

|  | (a) FAUST | (b) SCAPE | (c) FS_FAUST | (d) FS_SCAPE | (e) FS_SHREC'19 |

Figure 3: Performance evaluation on near-isometric shape matching. Models are trained and tested separately on FAUST and SCAPE, and cross-dataset generalization is assessed by testing on FAUST, SCAPE, and SHREC'19 using a combined FAUST+SCAPE training set. The figure shows Percentage of Correct Keypoints (PCK) curves with corresponding Area Under Curve (AUC) scores in the legend, offering a comparative view of matching accuracy and robustness across datasets.

Table 1: Quantitative results on near-isometric shape matching. Under certain conditions, our proposed method achieves superior performance compared to existing axiomatic, supervised, and unsupervised approaches.

| Train | FAUST | | | SCAPE | | | FAUST + SCAPE | | |
|---|---|---|---|---|---|---|---|---|---|
| Test | FAUST | SCAPE | SHREC'19 | FAUST | SCAPE | SHREC'19 | FAUST | SCAPE | SHREC'19 |
| **Axiomatic Methods** | | | | | | | | | |
| BCICP Ren et al. (2018) | 6.1 | 11.0 | – | 6.1 | 11.0 | – | 6.1 | 11.0 | – |
| ZoomOut Melzi et al. (2019b) | 6.1 | 7.5 | – | 6.1 | 7.5 | – | 6.1 | 7.5 | – |
| Smooth Shells Eisenberger et al. (2020) | 2.5 | 4.7 | – | 2.5 | 4.7 | – | 2.5 | 4.7 | – |
| DiscreteOp Ren et al. (2021) | 5.6 | 13.1 | – | 5.6 | 13.1 | – | 5.6 | 13.1 | – |
| **Supervised Methods** | | | | | | | | | |
| FMNet Litany et al. (2017) | 11.0 | 30.0 | – | 33.0 | 17.0 | – | – | – | – |
| 3D-CODED Groueix et al. (2018) | 2.5 | 31.0 | – | 33.0 | 31.0 | – | – | – | – |
| HSN Wiersma et al. (2020) | 3.3 | 25.4 | – | 16.7 | 3.5 | – | – | – | – |
| ACSCNN Li et al. (2020) | 2.7 | 8.4 | – | 6.0 | 3.2 | – | – | – | – |
| GeomFMaps Donati et al. (2020) | 2.6 | 3.4 | 9.9 | 3.0 | 3.0 | 12.2 | 2.6 | 2.9 | 7.9 |
| TransMatch Trappolini et al. (2021) | 1.7 | 30.4 | 14.5 | 15.5 | 12.0 | 37.5 | 1.6 | 11.7 | 10.9 |
| **Unsupervised Methods** | | | | | | | | | |
| DFAFMaps Luo et al. (2024) | 1.6 | 2.7 | – | 1.9 | 1.9 | – | – | – | – |
| SDUM Cao et al. (2024a) | 1.5 | – | – | – | 1.8 | – | – | – | 3.4 |
| RevisitingMap Cao et al. (2024b) | 1.5 | – | – | – | 1.8 | – | – | – | 3.4 |
| SSCDFM Sun et al. (2023) | 1.7 | – | – | – | 2.6 | – | – | – | 3.8 |
| DRecovery Sundararaman et al. (2024) | 1.5 | – | – | – | 1.9 | – | – | – | 4.8 |
| UnsupFMNet Halimi et al. (2019) | 10.0 | 29.0 | – | 22.0 | 16.0 | – | 11.0 | 13.0 | – |
| WSupFMNet Sharma & Ovsjanikov (2020) | 3.8 | 4.8 | – | 3.6 | 4.4 | – | 3.6 | 4.5 | – |
| SURFMNet Roufosse et al. (2019) | 15.0 | 32.0 | – | 32.0 | 12.0 | – | 33.0 | 29.0 | – |
| Deep Shells Eisenberger et al. (2020) | 1.7 | 5.4 | 27.4 | 2.7 | 2.5 | 23.4 | 1.6 | 2.4 | 21.1 |
| NeuroMorph Eisenberger et al. (2021) | 8.5 | 28.5 | 26.3 | 18.2 | 29.9 | 27.6 | 9.1 | 27.3 | 25.3 |
| ConsistFMaps Cao & Bernard (2022) | 1.5 | 3.2 | 19.7 | 3.2 | 2.0 | 28.3 | 1.7 | 3.2 | 17.8 |
| DUO-FMNet Donati et al. (2022) | 2.5 | 4.2 | 6.4 | 2.7 | 2.6 | 8.4 | 2.5 | 4.3 | 6.4 |
| AttentiveFMaps Li et al. (2022) | 1.9 | 2.6 | 6.4 | 2.2 | 2.2 | 9.9 | 1.9 | 2.3 | 5.8 |
| AttentiveFMaps-Fast Li et al. (2022) | 1.9 | 2.6 | 5.8 | 1.9 | 2.1 | 8.1 | 1.9 | 2.3 | 6.3 |
| ULRSSM Cao et al. (2023) | 1.6 | 6.6 | 7.2 | 4.6 | 1.9 | 7.7 | 1.6 | 2.1 | 4.6 |
| EOT Le et al. (2024) | 1.5 | 3.4 | 5.5 | **1.6** | 1.8 | 7.0 | 1.6 | 2.2 | 4.7 |
| Hybridfmap Bastian et al. (2024) | 1.5 | 4.2 | 5.9 | 2.2 | 1.8 | **6.7** | 1.5 | **2.0** | **3.4** |
| SFmaps Magnet & Ovsjanikov (2024) | 1.9 | **2.4** | **4.2** | 1.9 | 2.4 | 6.9 | 1.9 | 2.3 | 3.6 |
| AFMap Li et al. (2022) | 1.9 | 2.6 | 6.4 | 2.2 | 2.2 | 9.9 | 1.9 | 2.3 | 5.8 |
| SSLMSM Cao & Bernard (2023) | 2.0 | 7.0 | 9.1 | 2.7 | 3.1 | 8.4 | 1.9 | 4.3 | 6.2 |
| **Ours** | **1.4** | 8.5 | 5.4 | 10.0 | **1.8** | 7.0 | **1.4** | **2.0** | 5.3 |

## 4.2 NON-ISOMETRIC SHAPE MATCHING

**Datasets**. To systematically evaluate the performance of our method on challenging non-isometric shape matching tasks, we adopt the representative SMAL dataset Zuffi et al. (2017), which contains 49 quadruped animal models across eight categories. Following the split protocol of Donati et al. (2022), five categories are used as the training set and the remaining three categories as the test set, forming a partition of 29 training samples and 20 test samples. This split ensures that the training set contains no shapes similar to those in the test set, thereby presenting a substantial non-isometric matching challenge to the model.

**Results**. Our method demonstrates strong performance on the SMAL dataset, as summarized in Table 2 and illustrated in Figures 4a and 5a. In this highly challenging cross-category matching scenario, our unsupervised approach yields competitive results without relying on any supervision. Although certain supervised methods (e.g., RevisitingMap and Hybridfmap) achieve lower errors on the training set, with values of 3.6 and 3.3 respectively, our unsupervised method still attains an error

of 4.3, outperforming most existing mainstream algorithms. These results indicate that our method demonstrates robust generalization ability and practical applicability in handling shape matching tasks with complex structures and large category variations.

### 4.3 SHAPE MATCHING UNDER TOPOLOGICAL NOISE

**Datasets**. Mesh data acquired from real-world scanned scenes often exhibit compromised topology due to interpenetrations between different body parts. Such topological noise causes non-isometric deformations of the internal geometry of shapes, posing great challenges for shape matching algorithms. To evaluate the robustness of our method under such perturbations, we employ the TOPKIDS dataset Lähner et al. (2016). This dataset comprises several synthetic child shapes with artificially degraded topology through the merging of intersecting surface parts, resulting in deliberately constructed topological defects. The comparative experiments are conducted among unsupervised, supervised and axiomatic methods. Ground-truth correspondences are established by pairing all shapes with a selected reference shape.

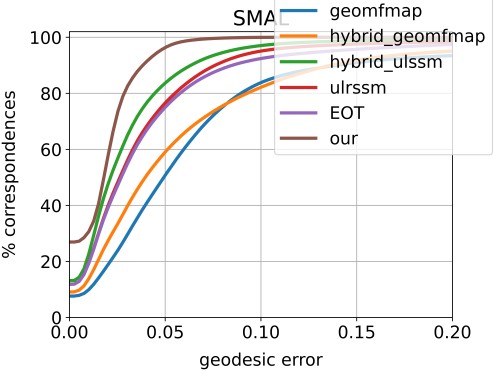 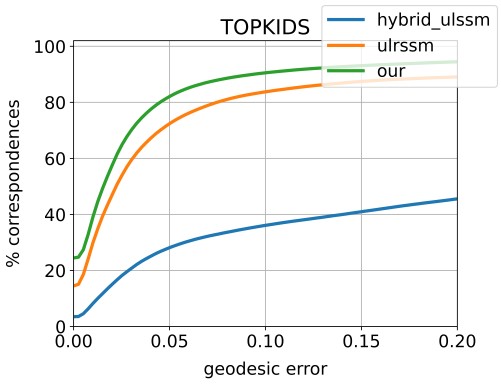

(a) Non-isometric matching results on the SMAL.     (b) Non-isometric matching results on TOPKIDS.

Figure 4: Matching results of our method on the non-isometric SMAL (a) and the topological noise TOPKIDS (b) datasets.

Table 3: Performance under topological noise on TOPKIDS. Our method demonstrates enhanced robustness against topological noise compared to existing approaches.

| Method | Geo. error ($\times$100) | Method | Geo. error ($\times$100) |
|---|---|---|---|
| ZoomOut Melzi et al. (2019b) | 33.7 | ConsistFMaps Cao & Bernard (2022) | 39.3 |
| Smooth Shells Eisenberger et al. (2020) | 11.8 | AttentiveFMaps Li et al. (2022) | 23.4 |
| DiscreteOp Ren et al. (2021) | 35.5 | MWP Deng et al. (2022) | 5.7 |
| UnsupFMNet Halimi et al. (2019) | 38.5 | Hybridfmap Bastian et al. (2024) | 5.0 |
| SURFMNet Roufosse et al. (2019) | 48.6 | ULRSSM Cao et al. (2023) | 9.2 |
| WSupFMNet Sharma & Ovsjanikov (2020) | 47.9 | SDUM Cao et al. (2024a) | 5.4 |
| Deep Shells Eisenberger et al. (2020) | 13.7 | **Ours** | **4.9** |
| NeuroMorph Eisenberger et al. (2021) | 13.8 | | |

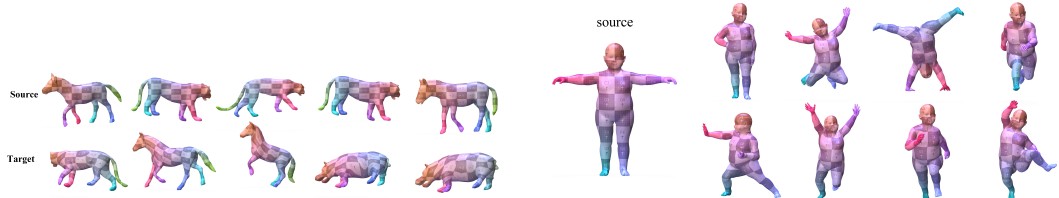

(a) Qualitative results on the SMAL datasets.     (b) Visual results on the TOPKIDS dataset.

Figure 5: Qualitative results on the (a) SMAL and (b) TOPKIDS datasets.

**Results**. As summarized in Table 3 and illustrated in Figures 4b and 5b, our method achieves strong performance in shape matching tasks under topological noise, exhibiting significantly superior robustness compared to existing functional map-based methods.

## 4.4 ABLATION STUDIES

To validate the effectiveness of our method, we conducted ablation studies on the SMAL dataset under a consistent evaluation protocol, where all models were assessed using weights trained for one epoch. Omitting the spectral mixture space (SMS) initially led to a high error of 11.7; reintroducing it significantly reduced the error, confirming its essential role in matching accuracy. Within the SMS framework, we compared three variants: (1) removing both structural attention and optimal transport (OT), using only a basic functional map pipeline with nearest-neighbor matching; (2) removing structural attention while retaining OT; and (3) removing OT while keeping structural attention. As shown in Table 4 and Fig. 6, removing any key component degraded performance, with errors rising from 4.3 (full model) to between 6.5 and 7.1. The highest error (7.1) occurred when both modules were omitted, indicating that structural attention and OT play complementary roles and jointly contribute to performance improvement.

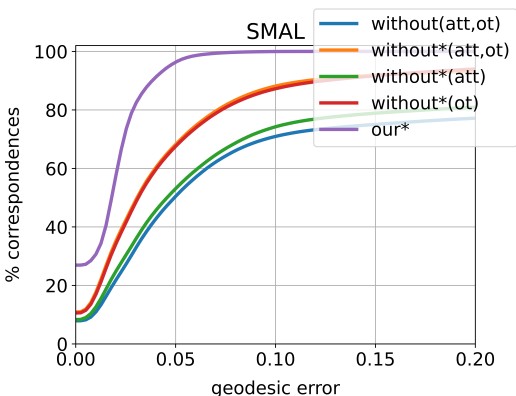

Figure 6: Ablation study on the SMAL dataset investigating the contribution of key components. The performance comparison demonstrates that integrating both dual-attention and optimal transport is essential for achieving high matching accuracy at low error thresholds.

## 5 CONCLUSION

This paper presents an unsupervised non-rigid shape matching framework that improves the discriminative capacity of shape features and matching accuracy by integrating structural information from hybrid spectral bases with local geometric features. A dual-level attention mechanism is introduced to enhance the feature extraction process, increasing its sensitivity to both geometric details and structural priors, which leads to more stable descriptors under complex deformations and partial occlusions. Additionally, a post-processing strategy based on Sinkhorn optimal transport is employed to optimize global consistency in initial point-to-point correspondences, effectively mitigating local mismatches and sparse mapping problems. Extensive experiments demonstrate that the proposed method achieves state-of-the-art performance on several challenging public datasets, exhibiting strong robustness in handling non-isometric deformations, topological perturbations, and partial observations.

Table 4: Ablation study on Geo. error ($\times 100$).

| SMS | ATT | OT | Geo. error |
|-----|-----|-----|-----|
| | | | 11.7 |
| ✓ | | | 7.1 |
| ✓ | | ✓ | 7.0 |
| ✓ | ✓ | | 6.5 |
| ✓ | ✓ | ✓ | 4.3 |

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
