# OpenReview forum: "Hybrid Dual-Attention Functional Map Learning for Robust Shape Matching"
_ICLR.cc/2026/Conference — Submitted to ICLR 2026_

### Official Review · Reviewer_3E7H · 2025-10-15

**Soundness:** 2
**Presentation:** 2
**Contribution:** 2
**Rating:** 2
**Confidence:** 5

**Summary:**

The paper proposes a hybrid dual-attention functional map learning framework for robust non-rigid 3D shape matching. Compared to existing deep functional map methods, the proposed approach introduces following modifications:
- It proposes a channel-wise attention and a vertex-wise cross-attention and self-attention mechanism to enhance the learned features from Siamese feature extractor.
- Instead of solely using LBO eigenfunctions as the bases for functional map computation, it uses additional elastic bases to compute a hybrid functional map.
- It introduces Sinkhorn optimization as an efficient optimal transport mechanism to refine point-to-point correspondences during inference.

In the experiment section, it shows that the proposed method achieves comparable results in comparison to state-of-the-art deep functional map methods and demonstrate that the proposed modifications improve the method performance in the ablation study.

**Strengths:**

1. In the experiment section, the paper makes an exhaustive comparison against most recent deep functional map methods and demonstrates comparable results for both near-isometric and non-isometric shape matching as well as state-of-the-art performance for shape matching under topological noise.

**Weaknesses:**

1. The methodological introduction of the proposed method is not very clear. For example, in line 228-230, it mentioned that the functional map C is used for the Sinkhorn optimal transport algorithm, while in Sec. 3.3 it does not indicate how the computed functional map is integrated in the SInkhorn optimization. In Sec. 3.4., the paper does not provide a definition of the loss function shown in Eq. 7.
2. The novelty of the proposed method is limited. For example, the use of hybrid functional map is already introduced in prior work (i.e. Hybridfmap Bastian et al. (2024)). The attention mechanism for feature extraction is also used in prior work [1] as well as the use of Sinkhorn optimization for point-wise correspondence refinement [2].

[1] Souhaib Attaiki, et al.: Dpfm: Deep partial functional maps (3DV 2021).

[2] Marvin Eisenberg, et al.: Deep shells: Unsupervised shape correspondence with optimal transport (CVPR 2020).

3. The performance improvement is marginal in both near-isometric and non-isometric settings. Moreover, the method seems to have worse cross-dataset generalization ability as shown in Tab. 1 for the case of FAUST-SCAPE and SCAPE-FAUST.

**Questions:**

1. It is interesting to see whether the proposed attention mechanism is robust to variable mesh sampling similar to DiffusionNet. Therefore, it is recommended to test the method on FAUST_a and SCAPE_a datasets as ULRSSM Cao et al. (2023) did.

---

### Official Review · Reviewer_26Zf · 2025-10-23

**Soundness:** 3
**Presentation:** 3
**Contribution:** 2
**Rating:** 4
**Confidence:** 4

**Summary:**

The paper proposes an unsupervised learning method for non-rigid shape matching. Based on the Deep Functional Maps framework, it proposes to: 1) combine LBO eigenfunctions and elastic basis and extract features with a dual-attention mechanism; 2) integrate an optimal transport mechanism to refine the output predictions. The method is tested on classical datasets (FAUST, SCAPE, SMAL, SHREC19, TOPKIDS), yielding results similar to those of the state of the art.

**Strengths:**

- The overall pipeline is sensible, and the explanation is straightforward. The design choices are validated by an ablation.
- Results on TOPKID with topological noise are compelling, with a high number of exact matches
- Comparison includes an extensive set of methods, serving as a good reference

**Weaknesses:**

- The contribution is mainly based on already known components, as a combination of elastic and LBO basis has already been seen in Hybrid FMap, and post-processing optimal transport has been adopted by Le et al. 2014. Hence, the main source of improvement appears to be the use of dual attention, which looks to be more of an engineering contribution. It would be great if such a technique would lead to some intuition about the nature of LBO and elastic basis combination, but the paper does not provide an analysis on this aspect, and hence it is difficult to understand the core take-home message underlying the proposed approach.

- The method's performance is generally in line with the state of the art, and I appreciate the results on topological noise. However, the paper does not offer results on more recent datasets, such as DT4D [1], which is nowadays commonly used in shape matching, nor on partial shapes [2]. On this latter point, I understand that the scope of the paper is focused on full-to-full cases, but testing the method on partial cases and even reporting failure cases would be highly informative for future work.

- The method seems to perform the best when training and test coincide (Tab. 1). It is worth highlighting that Hybrid FMAP performs slightly better, suggesting that at least for such classical full-shape datasets, the contribution provided by the method is rather incremental. The results on the combined training dataset (FAUST+SCAPE) seem to indicate that the increased capacity of the network is beneficial, since it does not lower the performance much on the two individual datasets and further decreases the error on SHREC19. However, Hybrid FMAP outperforms it on SHREC19, which again seems to indicate the method does not provide significant advances. Also, the paper does not report any analysis on the computational side, which I believe might be a drawback of relying on a more powerful backbone and the OT refinement.

[1]: Li, Y., Takehara, H., Taketomi, T., Zheng, B., & Nießner, M. (2021). 4dcomplete: Non-rigid motion estimation beyond the observable surface. In Proceedings of the IEEE/CVF International Conference on Computer Vision (pp. 12706-12716).

[2]: Ehm, V., Amrani, N. E., Xie, Y., Bastian, L., Gao, M., Wang, W., ... & Bernard, F. (2025, August). Beyond Complete Shapes: A Benchmark for Quantitative Evaluation of 3D Shape Surface Matching Algorithms. In Computer Graphics Forum (Vol. 44, No. 5, p. e70186).

**Questions:**

1) Could you comment on the contribution and provide an insight into why it leads to an improvement? What kind of problem does it solve?
2) I notice that in some cases the results differ from those of the original paper (e.g., hybrid FMAP reports 3.6 on SHREC19, but here it is reported 3.4). Are the methods retrained, or are the numbers in the table taken from the original papers? Could you comment on this discrepancy?
3) Ablation is carried out on SMAL, but the main application of the method has been on near-isometric cases (humans). Could you comment on this choice?
4) Do you believe the method could be adapted to work in the partial case?
5) Could you provide a comment on the computational aspects of the method, both at training and inference?

---

### Official Review · Reviewer_jHUd · 2025-10-28

**Soundness:** 2
**Presentation:** 1
**Contribution:** 1
**Rating:** 2
**Confidence:** 5

**Summary:**

This paper proposes a shape-matching pipeline that combines Laplace–Beltrami (LBO) and Elastic eigenmodes within a deep functional maps framework.
Based on an existing hydrid approach, the method augments pipeline with attention blocks and refines correspondences using optimal transport.
Experiments are reported on standard datasets with ablations.

**Strengths:**

The overall narrative is clear and the pipeline components are understandable in intent.

Evaluation is fair and standard, with ablation supporting design choices.

**Weaknesses:**

**Positioning and Related Works**

The paper lacks a proper **Related Works** section, making it hard to place the contribution among other methods. Closely related works using Attention for descriptor learning in deep functional maps (e.g. Attaiki et al. 2021, Li et al. 2022), or using optimal transport for shape matching, are **not cited**.

Le et al. (2024) is mentioned, but the authors don't highlight the differences. In particular, this paper solves a full dense Sinkhorn algorithm instead of its sliced version used in Le et al. (2024).

Several closest baselines are only introduced in the Results section and not mentioned earlier.

The relationship to Bastian et al. is underdeveloped. It is unclear what actually differs and what is adopted (see below for precisions). Space is instead taken by the description of the Elastic basis.

Datasets are cited via Cao et al. (2023) rather than citing the original dataset papers: this should be corrected.

**Clarity**

In Section 3.1, the definition of the Attention block is unclear, with no citation to SGCA. Key symbols are not defined (what are $L_x$, $\mu$ and $\sigma$ ?), the "structural guidance scaling factor" is not characterized (scalar ?), matrix shapes are inconsistent (e.g. $W_2$ shape is wrong). This whole section defines features $F$ that are **never referenced later**.

In Section 3.2, the complete mixed functional map is described in Eq (5). From the text alone, it seems the 4 blocks of this matrix are considered, where Bastian et al. (2024) discard the off-diagonal blocks. However, reading the rest of the paper (like caption of Figure 1), it seems these off-diagonal blocks are also discarded here. Furthermore, there is no mention of how this functional map is actually computed, except in the Figure 1.

In Section 3.3, optimal transport is defined using "LB spectral mapping" and "Elastic Modal Field mapping", **which are not defined**. Do these relate to the features $F$ ? What are the measures $u$ and $\nu$ ? The placement of this block in the pipeline (training, inference, ...) is therefore unclear. The section describes standard dense Sinkhorn without full details (marginal, epsilon, number of iterations), which aren't given later on either. There is also mention of high computational complexity, but this part is never discussed, while Le et al. (2024) specifically use sliced OT for this very problem.

Section 3.4 mentions the "isometric alignment" and "non-isometric detail alignment" losses, which are **never defined**. The hyperparameter $\alpha$ and the two schedules that both set alpha and "gradually introduce" the elastic basis are not described. It is unclear how the functional map is computed in this setting.


Generally, I'd say that the pipeline described is far from being reproducible with this amount of details.

**Results**

While sometimes competitive, results are **significantly** worse on some standard cross-dataset splits (8.5 on SCAPE or 10 on FAUST) , which contradicts strong SOTA claims.

Figure 3 is hard to parse. Terms like "hybrid_ulssm" and "hybrid_geomfmaps" are not explained in the text or the captions. Readers can't map curves to methods.

While the ablation is nice to have, it is surprising how poorly the naive baseline performs. I think there are several added blocks here which don't seem to bring much compared to simpler alternative.


**Style**

Citations should use `\citep` or `\citet` instead of being in line. Please move everything to present tense ("we proposed"), and ensure readability of the Figures.


**Overall**

While the idea is reasonable, the contribution appears very incremental, the exposition leaves key components underspecified, and the current description is insufficient for reproducibility.

**Questions:**

1 - Could you precisely define the Attention block ? What is $L_x$ ?

2 - Feature flow: How are the features $F$ used ?

3 - Are you using the off diagonal terms in Eq (5) ? What actually differs from Bastian et al. (2024) ?

4 - For optimal transport, you mention high cost. However your algorithm uses a simple dense Sinkhorn algorithm. Why not use some existing faster implementation using either dual potentials, sliced OT (as Le et al. (2024)), annealing, GPU specific implementations ... ? How do you compute the gradient of this result ?

5 - What losses do you use ? And what schedule for the hyperparameters ?

6 - What are "hybrid_ulssm" and "hybrid_geomfmaps" ?

---

### Official Review · Reviewer_AEzL · 2025-10-28

**Soundness:** 3
**Presentation:** 2
**Contribution:** 1
**Rating:** 2
**Confidence:** 3

**Summary:**

This paper proposes an unsupervised framework for 3D non-rigid shape matching using deep functional maps. The distinguishing points promoted are: 1. Use of 2 kinds of bases, the standard LBO and the elastic eigen basis. 2. an attention module in the feature extraction phase, and 3. use of OT as a refinement of the point-to-point correspondences. The method is tested widely, including FAUST, SCAPE, SHREC, TOPKIDS, and SMAL. Results are somewhat favorable.

**Strengths:**

- This paper ticks many boxes in terms of a shape-matching pipeline. The evaluation reported is wide, which is creditworthy.
- The results are favorable. Generally quite good for TOPKIDS and SMAL, and some are moderately comparable with SOTA in other cases

**Weaknesses:**

- The paper lacks a message. Individually - every component put together in this paper is quite well known in the shape matching community: use of elastic basis, optimal transport refinement, and attention in feature extraction.  The authors make no effort to expand a new concept through any of them.
- There is an overlooking at seemingly important details. Is Eq 2 correct? What is the difference between C and C_xy? How is eq2 a pointwise map recovery?
- The definition of the unsupervised loss is missing. What exactly is " isometric alignment loss"? how is the annealing done, and how robust is the strategy?
- The results from Tables 1 and 2 are competitive yet not outright impressive

**Questions:**

I credit the authors for the vast evaluation, including perhaps putting together a unique combination of components hitherto untried.  However, this paper feels unpolished and without a concrete message.

---

### Meta-Review · Area_Chair_YNrb · 2026-01-08

**Summary:**

The paper proposes an unsupervised framework for non‑rigid 3D shape matching within the deep functional maps paradigm. It combines Laplace–Beltrami and elastic bases, introduces a hybrid dual‑attention mechanism for feature extraction, and applies optimal transport refinement via Sinkhorn iterations. Experiments across multiple datasets (FAUST, SCAPE, SHREC, TOPKIDS, SMAL) show competitive results, with robustness under topological noise, but overall improvements are modest compared to prior work.

**Reviewer Concerns:**

Reviewers consistently raised concern that the contribution is insufficient and incremental, as each component has been explored in prior work. Methodological clarity is also a major issue, e.g., definitions of losses, equations, and attention blocks were incomplete or inconsistent. Related work was insufficiently cited, and positioning against baselines was weak. Results, while sometimes favorable, were not consistently superior to state‑of‑the‑art methods, and cross‑dataset generalization is limited. Missing evaluations on newer datasets and computational analysis were also presented.

**Reviewer Scores:**

The submission received ratings of 2, 2, 4, and 2. The authors did not provide responses to these concerns, therefore the manuscript cannot be accepted.

---

### Decision · Program_Chairs · 2026-01-26

Reject